

# Development of a gene doping detection method to detect overexpressed human follistatin using an adenovirus vector in mice

Koki Yanazawa[1], Takehito Sugasawa[2], Kai Aoki[2], Takuro Nakano[1], Yasushi Kawakami[2] and Kazuhiro Takekoshi[2]

[1] Graduate School of Comprehensive Human Sciences, University of Tsukuba, Tsukuba, Japan
[2] Laboratory of Clinical Examination/Sports Medicine, Division of Clinical Medicines, Faculty of Medicine, University of Tsukuba, Tsukuba, Japan

Corresponding author
Kazuhiro Takekoshi,
k-takemd@md.tsukuba.ac.jp

## ABSTRACT

**Background:** Gene doping is the misuse of genome editing and gene therapy technologies for the purpose of manipulating specific genes or gene functions in order to improve athletic performance. However, a non-invasive detection method for gene doping using recombinant adenoviral (rAdV) vectors containing human follistatin (*hFST*) genes (rAdV<*hFST*>) has not yet been developed. Therefore, the aim of this study was to develop a method to detect gene doping using rAdV<*hFST*>.
**Methods:** First, we generated rAdV<*hFST*> and evaluated the overexpression of the *hFST* gene, FST protein, and muscle protein synthesis signaling using cell lines. Next, rAdV<*hFST*> was injected intravenously or intramuscularly into mice, and whole blood was collected, and *hFST* and cytomegalovirus promoter (*CMVp*) gene fragments were detected using TaqMan-quantitative polymerase chain reaction (qPCR). Finally, to confirm the specificity of the primers and the TaqMan probes, samples from each experiment were pooled, amplified using TaqMan-qPCR, and sequenced using the Sanger sequencing.
**Results:** The expression of hFST and FST proteins and muscle protein synthesis signaling significantly increased in C2C12 cells. In long-term, transgene fragments could be detected until 4 days after intravenous injection and 3 days after intramuscular injection. Finally, the Sanger sequencing confirmed that the primers and TaqMan probe specifically amplified the gene sequence of interest.
**Conclusions:** These results indicate the possibility of detecting gene doping using rAdV<*hFST*> using TaqMan-qPCR in blood samples. This study may contribute to the development of detection methods for gene doping using rAdV<*hFST*>.

## INTRODUCTION

Gene therapy using gene editing technology is emerging as a new treatment strategy for various genetic and acquired diseases (*Wang et al., 2019*). Gene therapy drugs,

**Table 1 Types and relative numbers of the top seven clinically approved vectors used in gene therapy.**

| Vector | Gene therapy clinical trials | |
|---|---|---|
| | Number | % |
| Adenovirus | 573 | 17.5 |
| Retrovirus | 536 | 16.4 |
| Naked/Plasmid DNA | 482 | 14.7 |
| Lentivirus | 331 | 10.1 |
| Adeno-associated virus | 263 | 8.0 |
| Vaccinia virus | 197 | 6.0 |
| Lipofection | 125 | 3.8 |
| Others | 673 | 23.5 |
| Total | 3,180 | 100 |

such as Collategene® (AnGes. Inc, Osaka, Japan) (*Pharmaceuticals & Medical Devices Agency (PMDA), 2019*), Zolgensma® (AveXis. Inc, Bannockburn, IL, USA) (*Novartis, 2019*), and Kymriah® (Novartis. Inc., Basel, Switzerland) (*Novartis, 2018*), have been approved. Animal experiments and clinical trials are being conducted to develop gene therapy strategies against diseases such as Duchenne muscular dystrophy (*Duan, 2018*) and hemophilia B (*High & Anguela, 2016*; *Naso et al., 2017*). Thus, gene therapy is progressing rapidly and is expected to become common.

WADA defines gene doping as "The use of nucleic acids or nucleic acid analogues that may alter genome sequences and/or alter gene expression by any mechanism. This includes but is not limited to gene editing, gene silencing and gene transfer technologies" (*The World Anti-Doping Agency (WADA), 2021a*). With the rapid development of genetic engineering and gene therapy, the World Anti-Doping Agency (WADA) has strongly warned against gene doping. WADA has added gene doping to the prohibited list in 2003 and established a committee in 2004 to investigate the latest advances in the field of gene therapy and methods of detecting gene doping (*The World Anti-Doping Agency (WADA), 2015*). In January 2021, WADA has published laboratory guidelines for gene doping detection based on polymerase chain reaction (PCR) indicating the establishment of standardization methods is on its way (*The World Anti-Doping Agency (WADA), 2021b*).

In gene therapy, gene carriers called vectors are used. The vector data in Table 1 was taken from the Gene Therapy Clinical Trials Worldwide (*Journal of Gene Medicine, 2021*) website. Viral vectors are widely used, with recombinant adenovirus (rAdV) vectors being the most common and in use (*Liang, 2018*; *Xia et al., 2018*; *Zhang et al., 2018*). Moreover, they are safe because the possibility of integrating into the human genome is very low, and they are characterized by transient increases in the expression of the target gene, followed by rapid disappearance from the body (*Wold & Toth, 2013*; *Lee et al., 2017*). These features are favorable for athletes who intend to engage in gene doping. Hence, we focused on the rAdV vector, which can be misused for gene doping.

One gene that can be exploited for gene doping is the human follistatin gene (*hFST*). In humans, FST is produced in the liver and secreted into the bloodstream by stimuli such as exercise (*Hansen & Plomgaard, 2016*). The secreted FST is delivered to skeletal muscles throughout the body where it binds to activin A and myostatin, inhibiting their action and suppressing TGF-β signaling (*DePaolo, 1997*; *Patal, 1998*). As a result, it alleviates Smad3-dependent suppression of Akt/mTOR/p70S6K signaling and induces skeletal muscle hypertrophy (*Bodine et al., 2001*). FST is on the WADA prohibited list (*The World Anti-Doping Agency (WADA), 2020*), and there are concerns that it may be misused for gene doping.

To detect gene doping, it is necessary to detect fragments of the transgene or vector. In a previous study, quantitative polymerase chain reaction (qPCR) was shown to be the easiest and the most specific method for detecting gene fragments (*Sugasawa et al., 2019*). Therefore, in this study, we developed a qPCR detection method. qPCR mainly uses SYBR-Green or TaqMan polymerases for amplification. A previous study reported that TaqMan-qPCR has higher sensitivity and specificity than SYBR-Green-qPCR (*Aoki et al., 2020*). Therefore, we developed a detection method using TaqMan-qPCR.

Therefore, the purpose of this study was to develop a method using TaqMan-qPCR to detect gene doping of human FST gene using the rAdV vector.

## MATERIALS & METHODS

### Cloning of recombinant adenovirus vector containing the human follistatin gene

The following plasmids were used in this study: Gen EZ™ ORF clone hFST in pcDNA3.1 (+) (GenScript, Piscataway, NJ, USA), pENTR4 (Thermo Fisher Scientific, Waltham, MA, USA), and pAd/CMV/V5-DEST (Thermo Fisher Scientific, Waltham, MA, USA). HEK 293A cells (Thermo Fisher Scientific) were used to clone and amplify rAdV vectors. Gen EZ™ ORF clone FST in pcDNA3.1(+) was used as a template to amplify the *hFST* gene with 5′-*EcoRI* and 3′-*BamHI* restriction sites using PCR. PCR product was inserted into the pENTR4 plasmid using T4 ligase (Promega, 92 Madison, WI, USA). The sequences of inserted hFST genes in the pENTR4 plasmids were read using sanger sequencing and confirmed to be correct sequences. The following experiments were performed in accordance with a previous study (*Sugasawa et al., 2019*).

## CELL EXPERIMENTS

### Cell culture

HuH7 cells (RIKEN Bio Resource Center, Tsukuba, Japan) were cultured in Dulbecco's modified Eagle's medium (DMEM) (Thermo Fisher Scientific, Waltham, MA, USA) containing 10% fetal bovine serum (Wako, Osaka, Japan) and 1% penicillin/streptomycin (Wako), in an incubator maintained at 37 °C with 5% $CO_2$ gas. HuH7 cell line established from a highly differentiated human hepatocellular carcinoma. HuH7 cells were used to mimic the high expression of FST in the liver. Cells were seeded in 12-well plates at a density of $1.0 \times 10^5$ cells and six-well plates at a density of $2.5 \times 10^5$ cells. After the cells

reached 90% confluence, the medium was changed to serum-free DMEM, and the cells were infected by rAdV.

C2C12 cells (RIKEN Bio Resource Center, Tsukuba, Japan) were cultured under the same conditions. C2C12 is an immortalized mouse myoblast cell line. C2C12 cells were used to mimic the high expression of FST in the local muscle. After the cells reached 70–80% confluence, the medium was replaced with DMEM containing 2% horse serum and 1% penicillin/streptomycin (differentiation medium). Differentiation into myotubes was induced by culturing in the differentiation medium (changed every 2 days) for 7 days. After incubation, the medium was changed to serum-free DMEM, and the cells were infected by rAdV.

### rAdV infection

HuH7 and C2C12 seeded in 12-well plates (for total RNA/DNA extraction) were infected with $1.0 \times 10^{10}$ vp, and those seeded in six-well plates (for total protein extraction) were infected with $4.0 \times 10^{10}$ vp of rAdV<*hFST*>. A group with the same amount of phosphate-buffered saline (PBS) and a group infected with the same amount of rAdV<*mCherry*> were considered as controls. Total RNA/DNA extraction and total protein extraction were performed 4 days after infection in HuH7 and 3 days in C2C12 cells.

### Total RNA/DNA extraction

Total RNA was extracted from culture cells. Sepasol RNA I Super G (Nacalai Tesque, Kyoto, Japan) was used for the extraction, as per the manufacturer's protocol. After extraction, the concentration of RNA in each sample was estimated using a microspectrophotometer (NanoDrop 1,000; Thermo Fisher Scientific, Waltham, MA, USA), and 500 ng of total RNA from each sample was reverse transcribed using 5 × PrimeScript RT Master Mix (Takara Bio, Shiga, Japan) to cDNA. The cDNA was diluted with a four-fold amount of purified water.

Total DNA was extracted from cell culture samples. A phenol/chloroform/isoamyl alcohol solution (Nacalai Tesque, Kyoto, Japan) was used for extraction, as per the manufacturer's protocol. After extraction, total DNA concentration of each sample was adjusted to 10 ng/μL.

The synthesized cDNA was used to measure transgene expression, and total DNA was used to detect the transgene fragments using SYBR-Green-qPCR.

### Immunoblotting

Total proteins were extracted from the culture cells using lysis buffer (1% NP-40, 0.1% SDS, 20 mM Tris-HCl [pH 8.0], five mM ethylenediamine tetraacetic acid [EDTA], 150 mM NaCl, and proteinase inhibitor (Nacalai Tesque, Kyoto, Japan)). Lysates were centrifuged at 12,000×g for 15 min at 4 °C. Total protein concentration for each sample was measured using the BCA protein assay kit (Takara Bio, Shiga, Japan), and five μg/lane (HuH7) or 10 μg/lane (C2C12) of total protein was used for gradient gel electrophoresis. For western blotting, the blots were incubated with primary antibodies.

Table S1 lists the antibodies used in this study. Horseradish peroxidase-conjugated anti-rabbit IgG and anti-mouse IgG were used as the secondary antibodies. Signals were detected using a chemiluminescent reagent (EzWestLumi One; ATTO, Tokyo, Japan). Blots were scanned using a Light-Capture Cooled CCD Camera System (Image Quant LAS-4,000; GE Healthcare, Chicago, IL, USA).

## Immunofluorescence

Immunofluorescence was performed 3 days after rAdV<*hFST*> infection of C2C12 cells. After removing the supernatant, cells were fixed with 4% paraformaldehyde (Wako) for 15 min at 20 °C. Subsequently, the cells were permeabilized using 0.1% TritonX-100 (Nacalai Tesque) in PBS and blocked with 5% goat serum (Sigma-Aldrich, St. Louis, MO, USA) in PBS for 1 h at 20 °C. The primary antibody against Follistatin (Proteintech Group, Chicago, IL, USA, 60060-6-lg, 1:300) was diluted in 5% goat serum in PBS and incubated with the samples for 1 h at 20 °C. Wells were washed with PBS three times and incubated with the secondary antibody (Alexa Fluor 488 anti-mouse IgG; Jackson ImmunoResearch Laboratories, Inc., West Grove, PA, USA, 1:400) for 1 h at 20 °C. Finally, the wells were washed with PBS three times, stained with Dapi-Fluoromount-G® (SouthernBiotech, Birmingham, AL, USA), and subjected to fluorescence microscopy.

# ANIMAL EXPERIMENTS

## Animal

The animal experiments conducted in this study were approved by the Animal Experiment Committee of the University of Tsukuba (Approval Number: 20-378). A total of 6-week-old male ICR mice were purchased from Central Laboratories for Experimental Animals (Tokyo, Japan) and subjected to a 1-week acclimation period in a cage (Maximum of five mice in one cage). Mice were housed in an air-conditioned, pathogen-free animal room with a 12/12 h light/dark cycle. The mice were allowed to consume normal solid food and water *ad libitum*. At the beginning of the experiment, the mice weighed 34.1~37.4 g and were 7 weeks old. If weight loss was less than 20% during the experiment, or if there was obvious injury or illness, euthanasia was to be performed. There were no mice that corresponded to the above in this experiment. The sample size for each experiment was determined with previous study (*Sugasawa et al., 2019*).

## Development of a detection method for gene doping using rAdV<*hFST*>

### Long-term detection of transgene fragments for intravenous injection

Approximately, 50 μL of whole blood was collected from the tail vein of 10 untreated mice aged 7 weeks into a 1.5-mL tube containing 150 μL of EDTA-disodium salt (EDTA-2Na)/PBS mixture, which was used as a pre-injection sample. A total of 4 days later, $2.0 \times 10^{11}$ vp of rAdV<*hFST*> was injected through the orbital venous plexus. Thereafter, approximately 50 μL of whole blood was collected from the tail vein at 3, 6, and 12 h, and 1, 2, 3, 4, 5, 6, and 7 days after injection.

Total DNA was extracted using a phenol/chloroform/isoamyl alcohol solution (Nacalai tesque, Kyoto, Japan). After extraction, the concentration of DNA each sample was adjusted to 10 ng/μL, and DNA was subjected to TaqMan qPCR to detect *hFST* and *CMVp* gene fragments.

### Detection of transgene fragments in each specimen for intravenous injection

Fourteen untreated mice aged 7 weeks were randomly divided into two groups: group 1 (G1) ($n = 6$) and group 2 (G2) ($n = 8$). G1 was injected with 100 μL of PBS, and G2 was injected with 100 μL of $2.0 \times 10^{12}$ vp/mL rAdV<*hFST*> through the orbital venous plexus. A total of 6 h after injection, the mice were administered general anesthesia by isoflurane inhalation, followed by whole blood collection with EDTA-2Na. The mice were euthanized by cervical dislocation. Whole blood was centrifuged at 5,000 rpm for 10 min at 4 °C, and plasma and blood cell fractions were collected separately. Each sample was stored at −20 °C until the next analysis.

Total DNA was extracted from each sample using NucleoSpin® cfDNA XS (Macherey-Nagel, Düren, Germany) for plasma fractions and NucleoSpin® Blood (Macherey-Nagel, Düren, Germany) for blood cell fractions, as per the manufacturer's protocol. After extraction, the concentration of DNA each sample was adjusted to 10 ng/μL. The adjusted DNA was used to detect *hFST* and *CMVp* gene fragments using TaqMan qPCR.

### Determination of specificity of the primers and TaqMan probe

The DNA samples from each specimen (whole blood, plasma, and blood cells) mentioned in "Long-term detection of transgene fragments for intravenous injection" and "Detection of transgene fragments in each specimen for intravenous injection" were pooled in a 1.5-mL tube. The pooled DNA was subjected to TaqMan qPCR to detect the transgene fragment using the same primers and TaqMan probes as in "Long-term detection of transgene fragments for intravenous injection" and "Detection of transgene fragments in each specimen for intravenous injection". After the TaqMan qPCR, the amplified product was collected in a 1.5-mL tube and purified with NucleoSpin® Gel and PCR Clean-up (Macherey-Nagel). The purified and amplified products were sequence by an external vendor (FASMAC, Kanagawa, Japan) using the Sanger Sequence Analysis Service. The obtained data were analyzed using CLC Sequence Viewer ver. 8.0 (QIAGEN, Hilden, Germany) and BioEdit ver. 7.2.5 (developer: Tom Hall).

### Long term detection of transgene fragments for intramuscular injection

Approximately, 50 μL of whole blood was collected from the tail vein of six untreated mice aged 7 weeks into a 1.5-mL tube containing 150 μL of EDTA-2Na/PBS mixture, which was used as a pre-injection sample. A total of 3 days later, $2.5 \times 10^{12}$ vp/mL of rAdV<*hFST*> was injected at 30 μL into the tibialis anterior and at 50 μL into the gastrocnemius. Thereafter, approximately 50 μL of whole blood was collected from the tail vein at 3, 6, and 12 h, and 1, 1.5, 2, 2.5, 3, 3.5, 4, and 5 days after injection.

Total DNA was extracted using a phenol/chloroform/isoamyl alcohol solution. After extraction, the concentration of DNA in each sample was adjusted to 10 or 100 ng/μL, and DNA was subjected to TaqMan qPCR to detect *hFST* and *CMVp* gene fragments.

## PRIMER DESIGN AND QPCR

The sequences of primers and TaqMan probes are listed in Table S2. Primers and TaqMan probes were synthesized by Integrated DNA Technologies (Coralville, IA, USA).

SYBR® FAST qPCR Master Mix (KK 4602; Kapa Biosystems, Wilmington, MA, USA) reagent was used to perform SYBR-Green qPCR. The volume of the template solution, final concentration of the primer, negative control, and experimental conditions were taken from a previous study (*Aoki et al., 2020*).

PrimeTime Gene Expression Master Mix (Integrated DNA Technologies, Coralville, IA, USA) reagent was used to perform TaqMan-qPCR. The volume of template solution, final concentrations of primer and probe, negative control, and experimental conditions were taken from a previous study (*Aoki et al., 2020*). For standard curve for absolute quantification, 100 pg/μL Gen EZ$^{TM}$ ORF clone FST in pcDNA3.1(+) was used. The coefficient of determination ($R^2$) of the calibration curve was set to $R^2 > 0.98$.

Both SYBR-Green qPCR and TaqMan-qPCR samples were run in duplicates, and QuantStudio 5 Real-Time PCR Systems (Thermo Fisher Scientific, Waltham, MA, USA) were used.

## STATISTICAL ANALYSIS

GraphPad Prism version 7.04 (GraphPad, Inc., La Jolla, CA, USA) was used for statistical analysis of the data. First, the Shapiro-Wilk normality test was performed on all data to check the normality of the distributions. Then, data following a normal distribution were subjected to an unpaired t-test or one-way analysis of variance test, and the Tukey–Kramer method was used for the *post hoc* test. For non-normal distribution, Mann Whitney's U test and Benjamini, Krieger and Yekutiel's two-stage method were used. *p*-values less than 0.05 were considered statistically significant. Cell experiments data were indicated mean ± SD. Animal experiments data were indicated mean ± SEM.

## RESULTS

### *hFST* gene and FST protein were overexpressed in HuH7 and C2C12 cells

The expression of hFST and FST proteins in HuH7 and C2C12 cells significantly increased ($p < 0.001$) in the group infected with rAdV<*hFST*>, compared with the group infected with PBS and the group infected with rAdV<*mCherry*> (Figs. 1A, 1C, 2A, 2C, 2D, S1A, S2A). The number of hFST gene fragments and FST protein in the cell culture supernatant also increased significantly ($p < 0.001$) (Figs. 1B, 2B, S1B, S2B), and the phosphorylation levels of Akt and p70S6K significantly increased ($p < 0.05$) in C2C12 cells (Figs. 2E, 2F).
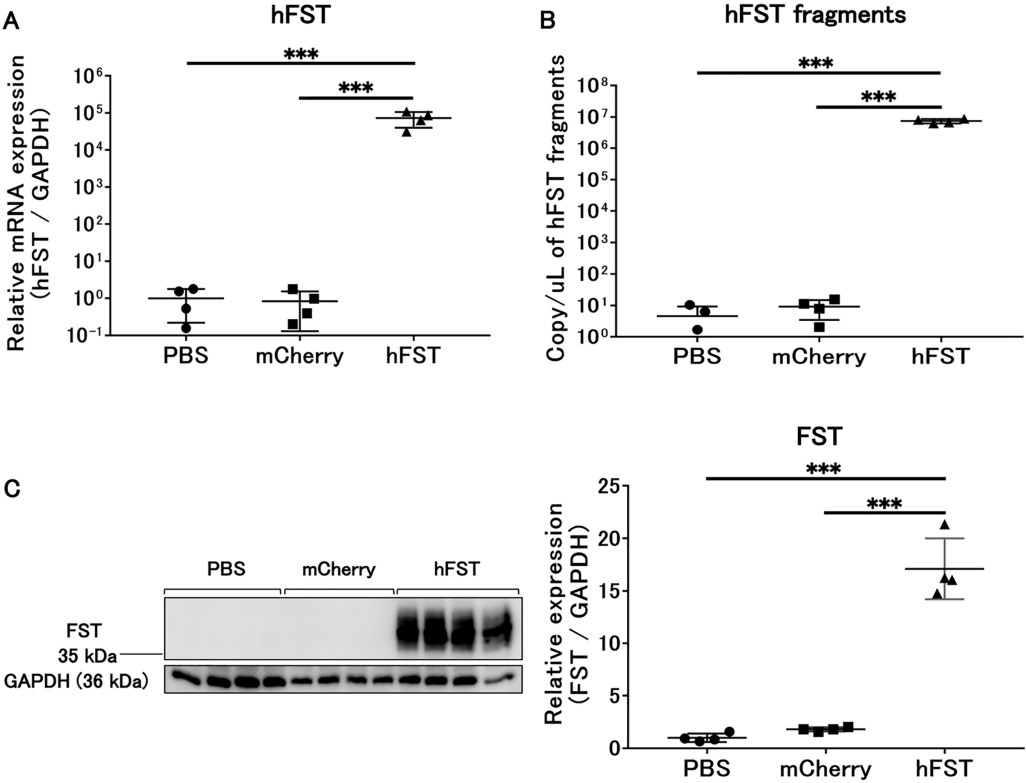

**Figure 1 Evaluation of gene and protein expression by rAdV<hFST> in HuH7.** (A) hFST gene expression. (B) Detection of hFST gene fragments in culture supernatant. (C) FST protein expression in cells. To confirm the rAdV was completely working, cell experiments were conducted. HuH7 cell line established from a highly differentiated human hepatocellular carcinoma. hFST gene and protein were overexpressed in hFST group. hFST gene fragment was detected from cell supernatant in hFST group. Data are means ± SD. ***$p < 0.001$.

## Transgene fragments were detected for 4 days after intravenous injection

Compared with the pre-infection samples, hFST transgene fragments were detected until 2 days ($p < 0.05$) and CMVp until 4 days ($p < 0.05$) after injection. In addition, both gene fragments were the highest at 6 h after injection (Fig. 3).

## Transgene fragments were detected in each specimen of intravenous injection

No transgene fragments were detected in any of the specimens in G1. In contrast, transgene fragments, including of hFST and CMVp, were detected in both specimens in G2. Moreover, when plasma and blood cell fractions were compared, transgene fragments were significantly more localized in the plasma fraction ($p < 0.05$) (Fig. 4).

## Primers and TaqMan probes were specific for the targeted transgene fragment

The results of the amplification curve of TaqMan-qPCR confirmed that the *hFST* and *CMVp* gene fragments in whole blood, plasma, and blood cell samples were fully amplified.

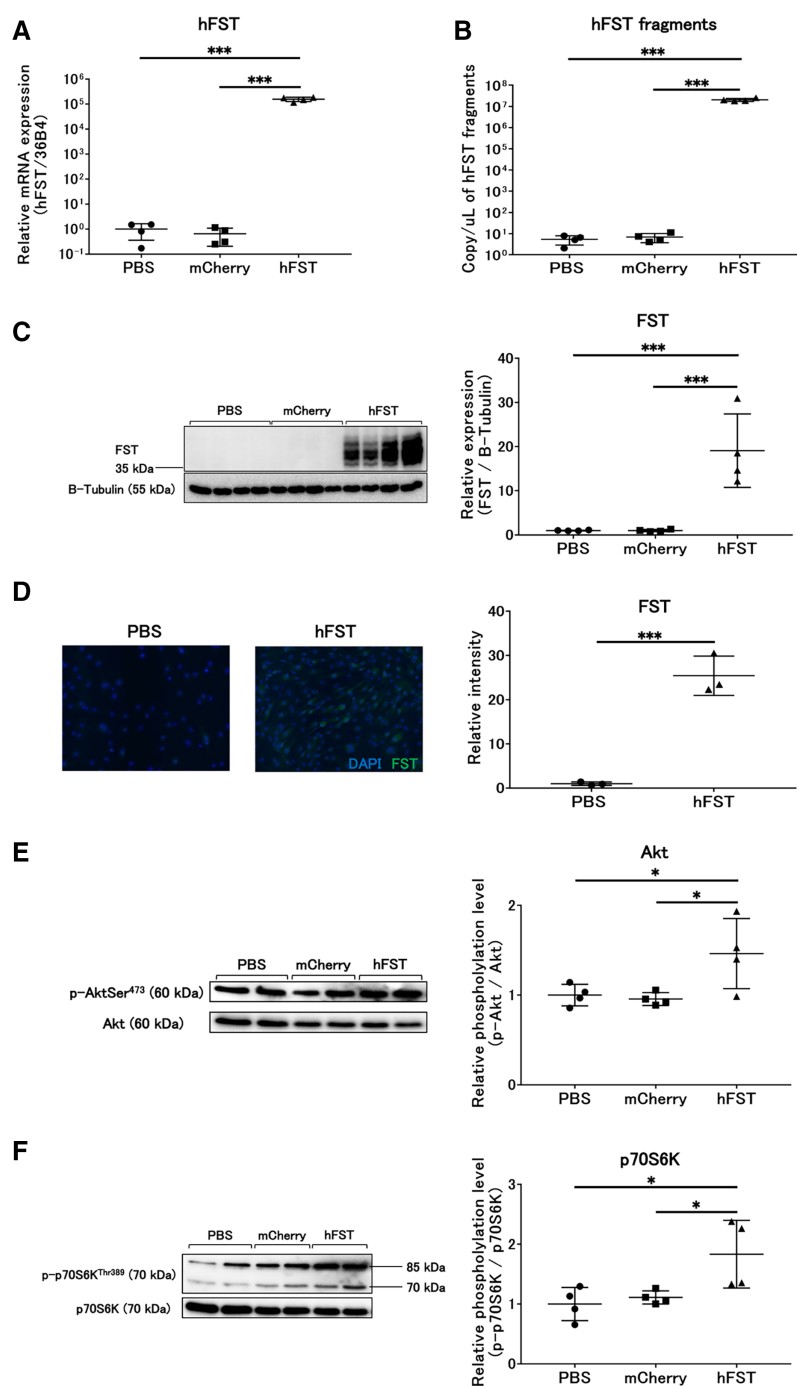

**Figure 2** **Evaluation of gene and protein expression and muscle protein synthesis signaling activity by rAdV<*hFST*> in C2C12.** (A) hFST gene expression. (B) Detection of hFST gene fragments in culture supernatant. (C) FST protein expression detected by western blot. (D) FST protein expression detected by immunofluorescence. (E) Akt phosphorylation level. (F) p70S6K phosphorylation level. To confirm the rAdV was completely working and muscle synthesis signals were activated, cell experiments were conducted. C2C12 is an immortalized mouse myoblast cell line. hFST gene and protein were overexpressed in hFST group. hFST gene fragment was detected from cell supernatant in hFST group. The phosphorylation levels of Akt and p70S6K significantly increased in hFST group. Data are means ± SD. $^*p < 0.05$ and $^{***}p < 0.001$.                         

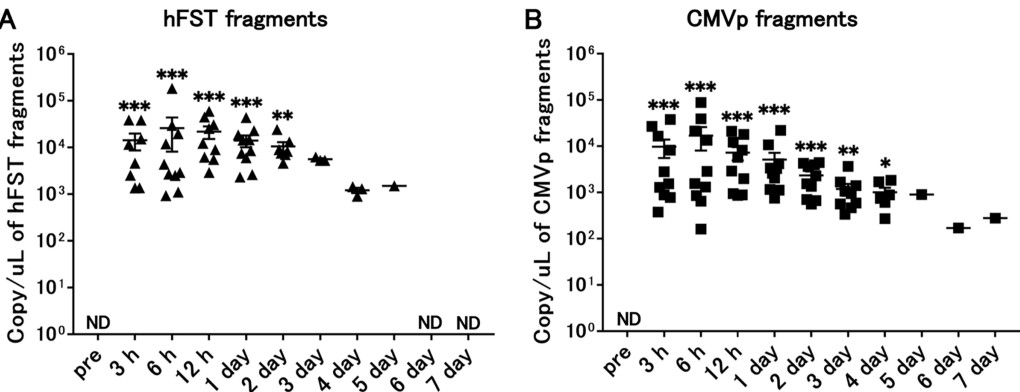

**Figure 3 Long-term detection of transgene fragments by intravenous injection.** (A) Detection of hFST gene fragments. (B) Detection of CMVp gene fragments. To confirm the detection period of the transgene fragments, animal experiments were conducted. rAdV was injected through the orbital venous plexus. hFST transgene fragments were detected until 2 days and CMVp until 4 days after injection. ND, not detected. Data are means ± SEM. $^*p < 0.05$, $^{**}p < 0.01$, and $^{***}p < 0.001$ *vs.* the pre-values before the injection.                  

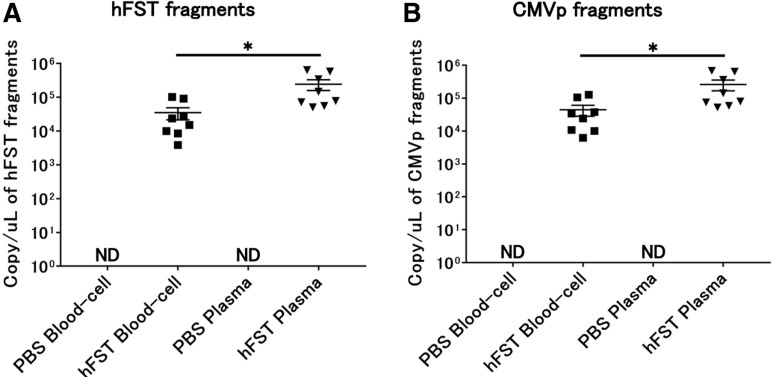

**Figure 4 Detection of transgene fragments from each sample by intravenous injection.** (A) Detection of hFST gene fragments in blood-cell and plasma fractions. (B) Detection of CMVp gene fragments in blood-cell and plasma fractions. To confirm the localization of the transgene fragment in the blood, animal experiments were conducted. hFST and CMVp were detected in plasma and blood cell fractions. Moreover, when plasma and blood cell fractions were compared, transgene fragments were significantly more localized in the plasma fraction. Data are means ± SEM. $^*p < 0.05$.
                  

Sanger sequencing showed that the sequences of all samples matched the reference sequences by more than 90 bases (Fig. 5). This confirmed that amplification by the prepared primers and TaqMan probe was specific.

## Transgene fragments were detected until 3 days after intramuscular injection

Compared with the pre-infection serum, after intramuscular injection, *hFST* transgene fragments were detected in 10 ng/μL of injected DNA until 12 h ($p < 0.05$) (Fig. 6A) and until 1 day ($p < 0.05$) for *CMVp* (Fig. 6C) and in 100 ng/μL of DNA until 2.5 days ($p < 0.05$) for *hFST* (Fig. 6B) and until 3 days ($p < 0.05$) for *CMVp* (Fig. 6D).

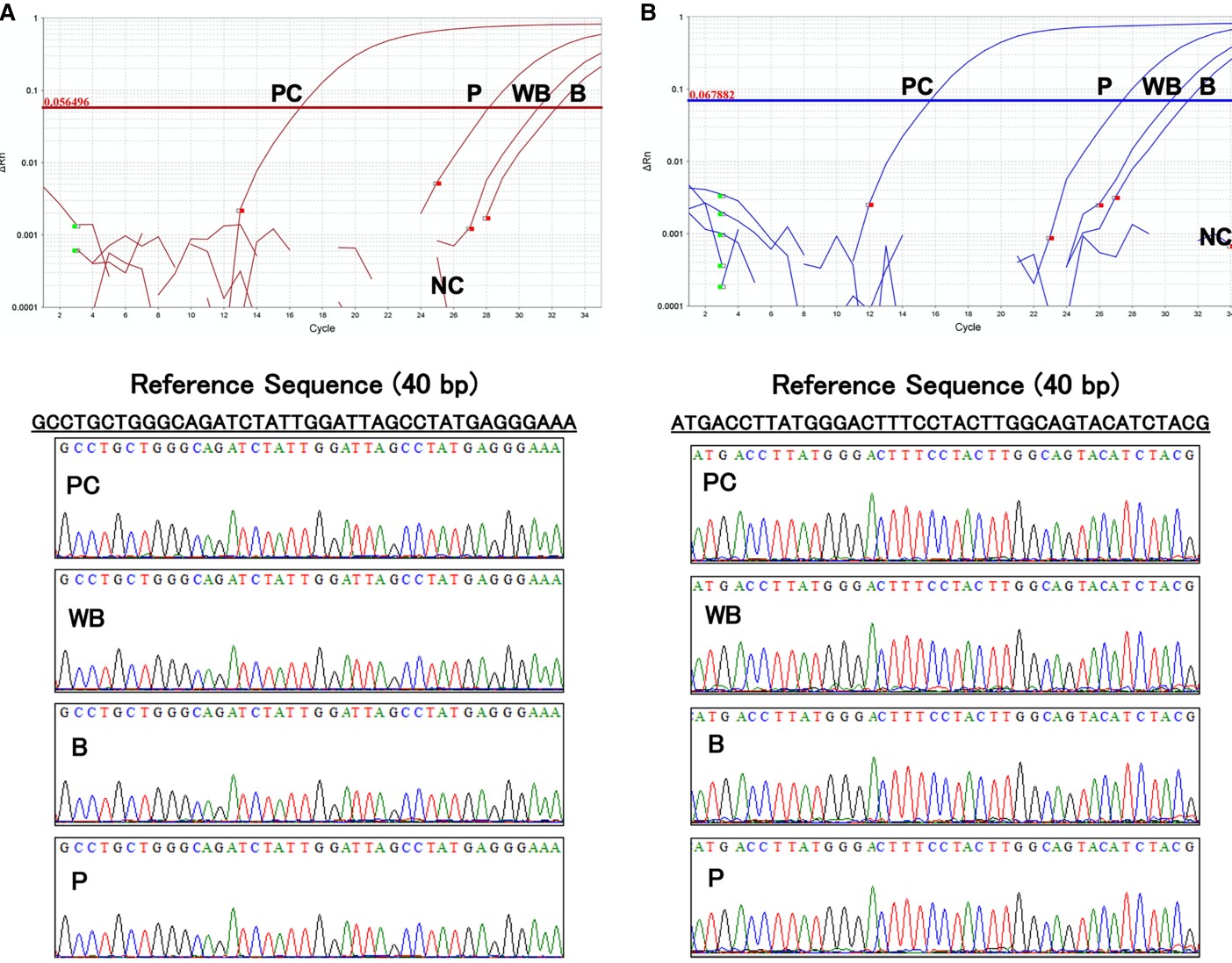

**Figure 5 Evaluation of the specificity of primers and TaqMan probes.** (A) Sequence of about 40 base pairs obtained from the amplification curve of the hFST gene fragments by TaqMan-qPCR and the waveform of the Sanger sequencing method. (B) Sequence of about 40 base pairs obtained from the amplification curve of the CMVp gene fragment by TaqMan-qPCR and the waveform of the Sanger sequencing method. PC means positive control (100 pg/uL pFST), WB, whole-blood DNA; B, blood-cell-fraction DNA, and P, plasma cfDNA. To confirm the specificity of the primers and TaqMan probe, Sanger Sequence Analysis was conducted. Primers and TaqMan probes were specific for the targeted transgene fragments.

## DISCUSSION

In this study, we evaluated gene and protein expression and muscle protein synthesis signaling using rAdV<*hFST*> and developed a detection method for gene doping by intravenous or intramuscular injection of rAdV<*hFST*>. In cell experiments, the *hFST* gene and FST protein expression in HuH7 and C2C12 cells significantly increased after injecting rAdV<*hFST*>, suggesting that the *hFST* gene and FST protein expression in the liver or muscle might significantly increase after intravenous or intramuscular injection of rAdV<*hFST*>. In addition, because FST is secreted extracellularly, the significantly raised FST in the liver or muscle would be secreted into the blood or near the muscle,

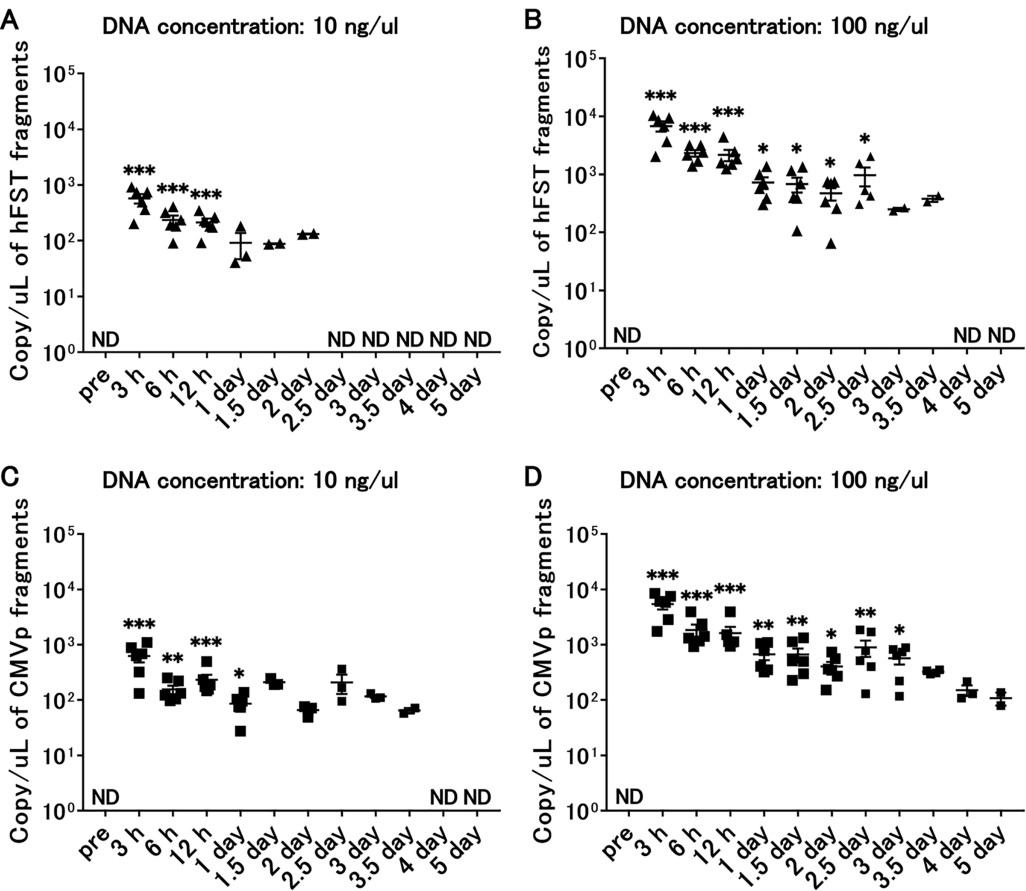

**Figure 6 Long-term detection of transgene fragments for intramuscular injection.** (A) Detection of hFST gene fragment using a sample with DNA concentration of 10 ng/uL. (B) Detection of hFST gene fragment using a sample with DNA concentration of 100 ng/uL. (C) Detection of CMVp gene fragment using a sample with DNA concentration of 10 ng/uL. (D) Detection of CMVp gene fragment using a sample with DNA concentration of 100 ng/uL. To confirm the detection period of the transgene fragments, animal experiments were conducted. rAdV was injected into the tibialis anterior and gastrocnemius. hFST transgene fragments were detected upon 2.5 days and upon 3 days for CMVp. ND, not detected. Data are means±SEM. *$p < 0.05$, **$p < 0.01$, and ***$p < 0.001$ *vs.* the pre-values before the injection.               

respectively. In addition, because the phosphorylation of Akt and p70S6K significantly increased in C2C12 cells, it is expected that muscle synthesis would have been activated to induce muscle hypertrophy. The results of the cell experiments in this study suggested that intravenous or intramuscular injection of rAdV<*hFST*> may be used to construct a gene-doping model.

We injected rAdV<*hFST*> in mice intravenously or intramuscularly and then sampled blood over time to see how many days after injection it took for the transgene and rAdV vector fragments to disappear. The results showed that gene fragments could be detected up to 4 days after intravenous injection (Fig. 3) and up to 3 days after intramuscular injection (Fig. 6). To date, no study has attempted to detect the *hFST* gene fragment, and these results are considered important findings for future research. However, further studies are needed to clarify the detectable period of rAdV<*hFST*> in

humans. It has been reported that drug metabolism in small experimental animals such as mice is about 10 times higher than that in humans (*Kato, 1981*), and it is possible that the rate of metabolism of genes introduced by adenovirus vectors differs between experimental animals and humans, but the extent of the difference is not yet clear. In addition, compared with the amount of virus used in humans (*Zhang et al., 2018*), this study used approximately 1/10 for intravenous injection and approximately 1/50 for intramuscular injection. Considering the body weight ratio (2,000:1) and circulating blood volume ratio (1,500:1) of humans (approximately 60 kg) and mice (approximately 30 g), a large amount of virus was administered in this study.

Doping tests require highly sensitive detection methods. To determine localization of the transgene fragments in whole blood, whole blood was collected and centrifuged into plasma and blood cell fractions and analyzed to determine which fraction contained the highest number of gene fragments. The *hFST* and *CMVp* gene fragments were localized more in the plasma fraction than in the blood cell fraction (Fig. 4). In contrast, in a previous study using rAdV, gene fragments were more localized in the blood cell fraction than in the plasma fraction (*Sugasawa et al., 2019*). This may be due to the time between injection and blood collection. Blood samples were collected 5 days after injection in the previous study, whereas blood was sampled 6 h after injection in this study. Because the injected rAdV vector was used as a viral solution, it is presumed to have localized in the plasma immediately after injection. It infected red blood cells or was phagocytosed by leukocytes over time. This is expected to lead to the development of more sensitive detection methods by determining the exact time points at which gene fragments localize predominantly in the plasma or in the blood cell fraction and which of the blood cell components (erythrocytes, leukocytes) or the surfaces of blood cells have the most localization.

Doping tests should have a high specificity. Therefore, it is necessary to develop a method to specifically detect target gene fragments. We used the Sanger sequencing method to verify the specificity of the detection performed in this study. As a result, we were able to specifically detect transgene fragments (Fig. 5). Furthermore, no non-specific amplification or contamination was found in the control group or samples before rAdV vector injection. These results indicate that the DNA extraction method, primer and TaqMan probe design, thermal cycling conditions, and reagents for TaqMan-qPCR in this study were accurate and optimal. Therefore, the protocol constructed in this study may be directly applicable to human specimens.

A limitation of this study was that we did not construct a gene doping model by intravenous or intramuscular injection of rAdV<*hFST*>. In other words, we were not able to establish a model that shows specific phenotypes of intravenous or intramuscular injection of rAdV<*hFST*>, such as changes in muscle strength, muscle wet weight, and muscle fiber type, owning to the increased levels of *hFST* gene and FST protein in the liver and muscle and activation of muscle synthesis. Therefore, it is necessary to establish a gene-doping model by intravenous or intramuscular injections in the future.

**Cell experiments**

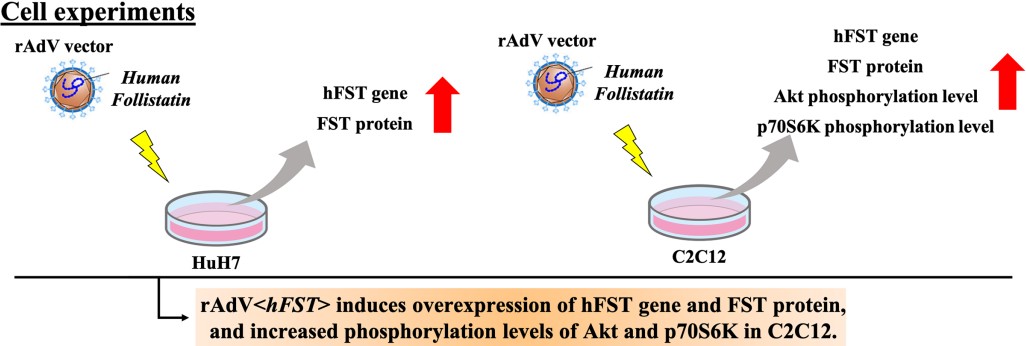

rAdV<*hFST*> induces overexpression of hFST gene and FST protein, and increased phosphorylation levels of Akt and p70S6K in C2C12.

**Animal experiments**

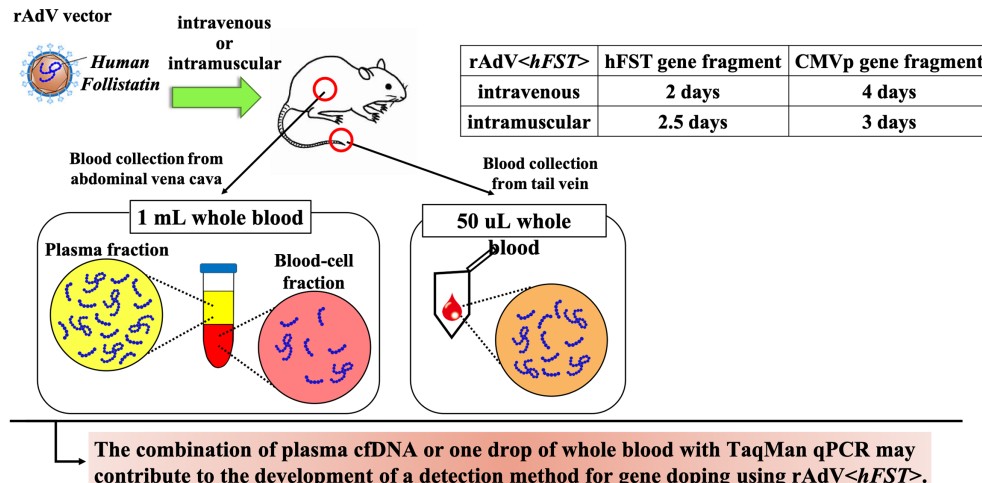

| rAdV<*hFST*> | hFST gene fragment | CMVp gene fragment |
|---|---|---|
| intravenous | 2 days | 4 days |
| intramuscular | 2.5 days | 3 days |

The combination of plasma cfDNA or one drop of whole blood with TaqMan qPCR may contribute to the development of a detection method for gene doping using rAdV<*hFST*>.

**Figure 7  Summary of this study.**

Research on methods for detecting gene doping has been conducted for only about 20 years, and it is still a developing research field with many unknowns. This study is a pioneering work in this field, and the results of this study may contribute to the development of detection methods for gene doping using rAdV<*hFST*>.

## CONCLUSIONS

In this study, we aimed to develop a method to detect multiple transgene fragments as proof of gene doping using rAdV<*hFST*>. Figure 7 presents a summary. This study showed that in the rAdV<*hFST*> transgenic model, multiple transgene fragments could be detected using TaqMan-qPCR from as little as 50 μL of whole blood, and each gene fragment was localized predominantly in the plasma. The new findings of this study may contribute to the development of detection methods for gene doping using rAdV<*hFST*>.

## ACKNOWLEDGEMENTS

We would like to thank Editage for English language editing.

### Funding
This work was supported by a Grant-in-Aid for Scientific Research KAKENHI from the Ministry of Education, Culture, Sports, Science, and Technology, Japan (No. 20H04062) and a grant from the promotional business of doping prevention activities, Japan Sports Agency (JSA). The funders had no role in study design, data collection and analysis, decision to publish, or preparation of the manuscript.

### Grant Disclosures
The following grant information was disclosed by the authors:
Ministry of Education, Culture, Sports, Science, and Technology, Japan: 20H04062.
Japan Sports Agency (JSA).

### Competing Interests
The authors declare that they have no competing interests.

### Author Contributions
- Koki Yanazawa conceived and designed the experiments, performed the experiments, analyzed the data, prepared figures and/or tables, authored or reviewed drafts of the paper, and approved the final draft.
- Takehito Sugasawa conceived and designed the experiments, analyzed the data, authored or reviewed drafts of the paper, and approved the final draft.
- Kai Aoki performed the experiments, analyzed the data, prepared figures and/or tables, authored or reviewed drafts of the paper, and approved the final draft.
- Takuro Nakano performed the experiments, authored or reviewed drafts of the paper, and approved the final draft.
- Yasushi Kawakami conceived and designed the experiments, authored or reviewed drafts of the paper, and approved the final draft.
- Kazuhiro Takekoshi conceived and designed the experiments, authored or reviewed drafts of the paper, and approved the final draft.

### Animal Ethics
The following information was supplied relating to ethical approvals (*i.e.*, approving body and any reference numbers):
The animal experiments conducted in this study were approved by the Animal Experiment Committee of the University of Tsukuba (Approval Number: 20-378).

### Data Availability
The raw measurements are available in the Supplementary File.

## Supplemental Information

Supplemental information for this article can be found online at http://dx.doi.org/10.7717/peerj.12285#supplemental-information.

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
