# Peer review of "Development of a gene doping detection method to detect overexpressed human follistatin using an adenovirus vector in mice"

_PeerJ, doi:10.7717/peerj.12285_

## Round 0.1 · original submission · Major Revisions

Please address the concerns of all reviewers.

Reviewer 1 ·

Basic reporting

- some essential literature is missing, referencing needs thorough review and upate (details given in the general reviewer comments section)

- the Figure quality / resolution needs to be improved (details given in the general reviewer comments section)

Experimental design

- the methods' description should be expanded to enable reproducing the experiments (details given in the general reviewer comments section)

Validity of the findings

no comments

Additional comments

Line 17-18: “Gene doping is a gene-editing technique to improve athletic performance without a target.”
 This introductory sentence should be phrased more precisely. Possible gene doping techniques comprise genome-editing techniques as well a gene therapeutic techniques without the potential of chromosomal integration. All are targeted techniques with the intention to manipulate specific genes or gene functions, respectively.

Line 48-50: “Gene doping is defined as "the non-therapeutic use of cells, genes, genetic elements or regulation of gene expression that have the ability to enhance athletic performance" (WADA, 2008).”
 Instead of citing the definition of the WADA prohibited list 2008, the reviewer suggests to cite the current version (WADA, 2021) as the definition has been updated and specified over the years (e.g. 2013, 2018).

Line 51-53: WADA has added gene doping to the prohibited list in 2003 and established a committee in 2004 to investigate the latest advances in the field of gene therapy and methods of detecting gene doping (WADA, 2016).

 Please provide the URL. Is it this one (from 2015)? http://antidopinglearninghub.org/sites/default/files/supporting-material/Anti-Doping%20Textbook%20-%202015%20Code.pdf

Line 54-55: However, standard methods to detect gene doping have not been established yet, and there is an urgent need to develop detection techniques.

 Several groups have already published successful detection approaches for potential gene doping targets via nested PCR, qPCR or ddPCR (e.g. Baoutina et. al. 2010; Beiter et. al., 2010; Ni et. al., 2011; Moser et. al., 2014, Neuberger et. al. 2016). In January 2021, WADA has published laboratory guidelines for gene doping detection based on polymerase chain reaction (PCR) indicating the establishment of standardization methods is on its way. The reviewer suggests that the authors elaborate/cite the current literature in more detail.

Line 56-60: In gene therapy, gene carriers called vectors are used. Data on vectors in Table 1 were
obtained from the Gene Therapy Clinical Trials Worldwide (Gene Therapy Clinical Trials
(2021)) website. Viral vectors are widely used, with recombinant adenovirus (rAdV) vectors
being the most common. They are also used in gene therapy (Liang, 2018; Xia et al., 2018; Zhang et al., 2018).

 “in gene therapy …viral verctors are used…they are also used in gene therapy…” The last sentence reiterates the two sentences before, please rephrase.

Line 86-88: The following plasmids were used in this study: Gen EZTM ORF clone hFST in pcDNA3.1(+) (GenScript, NJ, USA), pENTR4 (Thermo Fisher Scientific, MA, USA), and pAd/CMV/V5-DEST (Thermo Fisher Scientific).

 Please indicate if “Gen EZTM ORF clone hFST in pcDNA3.1(+)” is intronless.

Line 91-92 PCR product was inserted into the pENTR4 plasmid using T4 ligase (Promega software,
92 Madison, WI, USA).

 Do the authors mean „promega software“ or the enzyme from Promega? Please correct if so. Has the correct insertion of the hFST sequence confirmed by Sanger Sequencing? Please state here if so.

Line 97 ff: “2.1. Cell Culture…”

 Please add supplier information of cell culture media and antibiotics. Please add more information for the reader about the kind (and reason) of cell lines used in the experiments.

Line 178: Total DNA was extracted using a phenol/chloroform/isoamyl alcohol solution.

 Please add supplier information.


Line 218: The sequences of primers and TaqMan probes are listed in Table S2.

 Please provide sequence accession numbers used for primer designs and indicate primers used for SYBR Green assays or Taq Man Assays. Please provide information about the detection of hFST isoforms as well as on possible mice or human gDNA detection via the assays in the text document. Please give a description of primer location (e.g. exon 4-exon 5 spanning). Please indicate in which assay / figure mFST primer assay has been used.

Line 287-289 In addition, because the phosphorylation of Akt and p70S6K significantly increased, it is expected that muscle synthesis would have been activated to induce muscle hypertrophy.

 Phosphorylation of Akt and p70S6K has been shown in the mouse cell line modell (C2C12) but not in the human cell line (HuH7). Assumptions concerning functional effects via induction of muscle synthesis should be discussed more carefully and differentiated between species.

Line 325-327 These results indicate that the DNA extraction method, primer and TaqMan probe design, thermal cycling conditions, and reagents for TaqMan-qPCR in this study were accurate and optimal.

 Results show specific amplification of the intended target. Please provide further information about assay efficiency and LOD to substantiate “optimal” set up of the assay and cycling conditions.

Line 343 -344 In this study, we aimed to develop a method to detect multiple transgene fragments as proof of gene doping using rAdV<hFST>.

 Multiple transgene fragments? just hFST under CMVp?


Figure Legends

 In general, figure legends (e.g. “Figure 1 (A) hFST gene expression”) should provide more information for the reader about the experimental set up and shown results.

Figure 1 A) and B)
The reviewer recommends to provide SYBR Green melting curve profiles of qPCR assays in the supplements as well as agarose gel separation profiles to underline specific target amplification of the assays and proof of amplicon sizes.

Figure 5 A) and B)

 Please provide amplification plots in higher resolution and indicate NTCs.

Reviewer 2 ·

Basic reporting

Clear and unambiguous, professional English used throughout.
==>
Before final submitting, the manuscript should be edited by natives or the editorial office.

Literature references, sufficient field background/context provided.
==>
Yes

Professional article structure, figures, tables. Raw data shared.
==>
Yes

Self-contained with relevant results to hypotheses.
==>
Yes

Experimental design

Original primary research within Aims and Scope of the journal.
==>
Yes

Research question well defined, relevant & meaningful. It is stated how research fills an identified knowledge gap.
==>
Yes

Rigorous investigation performed to a high technical & ethical standard.
==>
Yes

Methods described with sufficient detail & information to replicate.
==>
Yes

Validity of the findings

Impact and novelty not assessed. Meaningful replication encouraged where rationale & benefit to literature is clearly stated.
==>
No problem.

All underlying data have been provided; they are robust, statistically sound, & controlled.
==>
Yes

Conclusions are well stated, linked to original research question & limited to supporting results.
==>
Yes

Speculation is welcome, but should be identified as such.
==>
No problem.

Additional comments

Recently, WADA published PCR-based detection guidelines. Does the detection method developed in this study follow this guideline?

https://www.wada-ama.org/sites/default/files/resources/files/wada_guidelines_for_gene_doping_pcr_test_v1_jan_2021_eng.pdf

Although the authors studied using a mouse model, the authors may need to follow the guideline to develop gene doping tests in sports. If there is a reason not to follow, the authors should consider the reason in the texts.

Primer, probe, sequencing
In this study, hFST was administered into mice and detected by PCR. After administering hFST to humans, can this be specifically detected the hFST with the primers and probes designed?
Sequencing may be difficult because of amplifying human genome .


Minor:
Line 17-18: "without a target". What does this mean?
Line 17-18: The authors mentioned "Gene doping is a gene-editing technique to improve athletic performance", but does "adenoviral (rAdV) vectors containing human follistatin (hFST) genes" correspond to gene editing?
Line 48-53: WADA defines gene doping as "The use of nucleic acids or nucleic acid analogues that may alter genome sequences and/ or alter gene expression by any mechanism. This includes but is not limited to gene editing, gene silencing and gene transfer technologies."
Line 54-55: The authors should cite the WADA guideline here.

Reviewer 3 ·

Basic reporting

The manuscritp is clear and the English language editing was done. There is one point that could be re-evaluated in the tile that is the term "adenovirus vector", because the focus in the title is gene doping method instead the method for overexpression. I believe that the result section should be more explored and the figure legend must be more detailed. Moreover, the conclusion section repets results and aim that must be avoided.

Experimental design

The authors did not say the number of independent experiments to validate the results. I believe that only one independent experiment was done based on raw data file.
The major concern is the capacity of adenovirus system to reach muscle cells or liver after infusion promoting the overexpression of FST over there, the current data do not bring this information.
The detection of hFst by PCR in the peripheral blood had as template the compounds of adenovirus available systemicaly, then how do the authors detect the gene doping after long time when adenovirus is gone and the target cells were modified. Also, how do the authors address the gene doping using integrated DNA or infusion of RNA that will not be readly available in the blood.

Validity of the findings

Some informations already said in the topic 2.

Additional comments

Dear auhtors,
The manuscript is interesting and the overexpression of FST was clearly demonstrated. I feel that the authors worked hard in the overexpression to simulate a gene doping, whereas the detection of gene doping is not a new method. I know that the protocol for detection of gene in the peripheral blood is an advantage developed by authors, but qPCR is already previously proposed to detect gene editing. So far, I believe that authors should demonstrate that the overexpression of FST really occured, in vivo, in the target cells.

---

## Round 0.2 · accepted · Accept

All critiques were addressed and I am happy to accept your revised manuscript now.

Reviewer 3 ·

Basic reporting

The authors suggest a methodology to detect gene doping using adenovirus vector and the gene used to demonstrate the strategy evidenced the application of the proposal. The authors answered my comments that will be evaluated properly in the future studies.

Experimental design

This point was evaluated previously.

Validity of the findings

This point was evaluated previously.

Additional comments

No comments.